# Potential of Low Cost Agro-Industrial Wastes as a Natural Antioxidant on Carcinogenic Acrylamide Formation in Potato Fried Chips

**DOI:** 10.3390/molecules27217516

**Published:** 2022-11-03

**Authors:** Adel Abdelrazek Abdelazim Mohdaly, Mohamed H. H. Roby, Seham Ahmed Rabea Sultan, Eberhard Groß, Iryna Smetanska

**Affiliations:** 1Department of Food Science and Technology, Faculty of Agriculture, Fayoum University, Fayoum 63514, Egypt; 2Department of Plant Food Processing, Agricultural Faculty, University of Applied Sciences Weihenstephan-Triesdorf, Markgrafenstr 16, 91746 Weidenbach, Germany

**Keywords:** acrylamide, agricultural wastes, antioxidant activity, phenolic compounds, potato chips, reduction

## Abstract

Acrylamide is classified as a toxic and a prospective carcinogen to humans, and it is formed during thermal process via Maillard reaction. In order to find innovative ways to diminish acrylamide formation in potato chips, several extracts of agricultural wastes including potato peels, olive leaves, lemon peels and pomegranate peels extracts were examined as a soaking pre-treatment before frying step. Total phenolic, total flavonoids, antioxidant activity, and the reduction in sugar and asparagine contents were additionally performed. Proximate composition of these wastes was found to be markedly higher in fat, carbohydrate and ash contents. Lemon peels and potato peels showed almost similar phenolic content (162 ± 0.93 and 157 ± 0.88 mg GAE /g, respectively) and exhibited strong ABTS and DPPH radical scavenging activities than the other wastes. The reduction percentage of reducing sugars and asparagine after soaking treatment ranged from 28.70 to 39.57% and from 22.71 to 29.55%, respectively. HPLC results showed higher level of acrylamide formation in control sample (104.94 mg/kg) and by using the wastes extracts of lemon peels, potato peels, olive leaves, and pomegranate peels succeeded to mitigate acrylamide level by 86.11%, 69.66%, 34.03%, and 11.08%, respectively. Thus, it can be concluded that the soaking of potato slices in the tested wastes extracts as antioxidant as pre-treatment before frying reduces the formation of acrylamide and in this way, the risks connected to acrylamide consumption could be regulated and managed.

## 1. Introduction

Food processing generally entails many requisite preparations that are applied to transform the raw materials into food or food into consumable forms. Many different processing steps, including washing and cutting, frying, drying, roasting, grilling, fermenting and cooking are involved as a single process or a combination. Throughout these procedures, some undesirable substances that have negative nutritional effects and potentially dangerous consequences on human health can be produced [1]. Acrylamide (AA) is one among such heat-induced carcinogenic compound that could occur in starchy rich foods when they are processed at high temperatures (>120 °C) for at least 20 min, but it is not present in unheated food [2]. It is a low molecular weight, crystalline substance that is colorless, odorless and stable at ambient temperature [3]. It is highly soluble in polar solvents such as water [4] and organic solvents with variable degrees of solubility [5]. Acrylamide, once recognized as a crucial component for industries because of its versatility, poses a risk to the safety of many processed food products [3]. Acrylamide, besides being a recognized neurotoxin, has been identified as a potential carcinogen to humans by the World Health Organization (WHO) and high quantities of AA have been found in several commonly consumed food items. Recent research has demonstrated that acrylamide possess immunotoxic, genotoxic, neurotoxic, mutagenic, and carcinogenic effects on animals [6]. The maximum daily dose of acrylamide that should be given is 10 μg /kg of body weight [7]. Maillard reaction, a kind of nonenzymatic browning reaction, contributes to the formation of various important features of food quality, including color, texture, flavor and aroma formation in many cooked foods subjected to baking, roasting or frying. However, the main pathway of AA formation during thermal processing is Maillard reaction between reducing sugar and amino acids, mostly asparagine [8]. Over the years, scientists and food manufacturers worldwide have made many efforts to achieve AA reduction in various processed foods, especially potato-based processed products such as crisps and chips, which are a major contributor to acrylamide daily intake [9]. In view of this, several methods have been proposed in order to reduce the level of acrylamide produced during food preparation and processing such as breeding and selecting potatoes varieties with low reducing sugar contents, altering the processing parameters such as reduction in temperature or time of cooking, and controlling of storage conditions. The pre-processing method before frying such as blanching, addition of salts, soaking, addition of vinegar, organic acids, and antioxidants has also been discovered for acrylamide mitigation. Use of enzymes, such asparaginase, to prevent the formation of acrylamide through the enzymatic conversion of L-asparagine to ammonia and L-aspartic acid has also been observed as a mitigation technique [10]. However, even when produced from diffused and renewable sources, the price of enzymes remains expensive [11].

Numerous agricultural wastes represent a great source of bioactive compounds which are useful not only in food industries but also in various industrial sectors such as cosmetics, adhesives, pharmaceuticals, and nutraceuticals [12]. Potato (*Solanum tuberosum*) is the fourth most important agricultural crop in the worldwide after wheat, maize, and rice [13]. The processing of raw tubers, the edible part of the plant, commonly involves peeling that produces a lot of bulky waste that is either discarded or utilized as animal feed, whose disposal creates serious environmental concerns. Processed potato products such as hash browns, puree, chips and frozen food amount of waste produced by potato peeling varies between 15% and 40%, based on the peeling method. Moreover, potato starch and canning industries also produce a great volumes of peel wastes with accompanying environmental consequences [14]. Annually, these industries contribute around 70 to 140 thousand tons of potato peels worldwide [15]. Olive wastes such as olive leaves are produced in large amount during pruning of the olive tree and do not have any practical applications. However, they have been reported to have various phytochemical compounds and, particularly, they are a potential source of phenolic [16]. Oleuropein is the main phenolic substance of olive leaves. Oleuropein has a diversity of pharmacological characteristics such as anti-atherogenic, anti-cancerogenic, antimicrobial, antioxidant, anti-inflammatory, and antiviral activities [17]. Lemon (*Citrus lemon*) is one of the most widely consumed citrus fruits in the worldwide, which is used for therapeutic, nutritional and beauty purposes. The production of lemon juice produces around 20–30% juice, leaving 50–60% discarded as a waste, consisting mostly of lemon peel. Thus, a considerable quantity of lemon peel is generated worldwide each year. This waste, which is both expensive and harmful to the environment, must be disposed of. There are numerous uses of lemon peel processes, such as pectin extraction, livestock feed, and essential oil extraction [18]. Lemon peel contains a wide range of beneficial bioactive compounds that are great for improving health with substantial antioxidant activity. Pomegranate (*Punica granatum* L.) is one of the most significant fruit crops in arid and semiarid areas. Pomegranate peels are often created after processing pomegranate fruits (pomegranate juice processing), which makes up about 40–50%, has been frequently regarded as an agricultural waste product. The pomegranate peel is a rich source of bioactive phenolic and the largest classes include flavonoids, anthocyanins and tannins. These compounds are responsible for its higher antioxidants, antibacterial, antifungal, and anti-diabetic properties [19]. However, many reports have already revealed that these agricultural wastes are a useful source of health-promoting phytochemicals exhibiting antioxidant activities, however, on the contrary, due to their variable quantities of phenolic acids, whose effect on the formation of acrylamide is doubtful, their natural extracts could be potential sources to reduce acrylamide formation in potato-based products. Therefore, our research topic is the effect of natural wastes such as potato peels, olive leaves, lemon peels and pomegranate peels extract as a soaking pre-treatment method on the mitigation of AA formation in processed potato product after frying process. The main goals of this study were: (i) to assess acrylamide contents in potato chips with the tested samples; (ii) to assess reducing sugar and asparagine levels in the produced potato chips samples and correlate them with acrylamide levels; (iii) to determine total phenolic and total flavonoids contents, HPLC analysis of phenolic compounds, and antioxidant scavenging activity of tested samples.

## 2. Results and Discussion

### 2.1. Chemical Composition of Raw Materials

Table 1 presents the chemical composition of lemon peels, potato peels, olive leaves, and pomegranate peels. From the data, it was observed that the composition analysis of lemon peels showed 73% moisture, 3.70% crude fat, 1.89% crude protein, 7.42% ash, and 13.99% carbohydrates. The lipid content in lemon peels was lower than that reported by Janati et al. [20]. Meanwhile, ash content of lemon peels in this work was 7.42% in accordance with Janati et al. [20] who reported the value of ash was 6.26%. The most abundant macronutrients in olive leaves, lemon peels, potato peels, and pomegranate peels are carbohydrates, presenting 28.67, 13.99, 9.26 and 19.05 g/100 g FW, respectively, confirmed with the study by Shirley et al. [21]. Regarding crude protein, the total content in all samples is low, ranging around 1.41–2.33%. Our results in potato peels are in agreement with Javed et al. [22] in which the chemical composition of different variety of raw potato peels was 83.3–85.1, 1.2–2.3, 0.1–0.4, 8.7–12.4, and 0.9–1.6 for moisture, protein, lipids, carbohydrate and ash contents, respectively. On the other hand, pomegranate peels contained moisture (74%), crude fat (2.50%), crude protein (1.41%), ash (3.03%) and carbohydrates (19.05%). Our results are supported by a previous research of Kyriakos et al. [23] who found that pomegranate peels are rich in carbohydrates and ash, whereas contain a slight amounts of crude protein. So, with the value of ash reported in this work, all wastes samples might be suitable for use in animal feeds. On the basis of the above, it can be concluded that these wastes could provide as a supply of fat, carbohydrate and ash.

### 2.2. Extraction Yield

Solvent extraction is a widely applicable and the most commonly used technique in isolation and extracting of bioactive compounds. Aqueous methanol was chosen as an ideal extraction solvent for extracting polyphenolic and bioactive compounds [24]. The extraction amount obtained of wastes materials are displayed in Table 1. The aqueous methanol extraction reveals varying levels of extractable soluble compounds among potato peels, lemon peels, olive leaves, and pomegranate peels. Pomegranate peels had the greatest extraction yield (63.29%), about four times more than potato peels (14.50%) and two times more than olive leaves (32.98%) while lemon peels showed 24.85% of extracts. Our work demonstrated a high level of extraction yield compared to other studies. Azadeh et al. [25] found that water produced the highest amount of potato peel extract (11.2%), followed by methanol (8.03%) and ethanol (5.6%). In research by Shalini et al. [26], the highest extraction yield of pomegranate peels (16.28%) and the lowest amounts (5.74%) were obtained from ethanol: water (1:1) and water, respectively. Moreover, Salah et al. [27] found the yield of extracts of olive leaves, obtained from different varieties using a mixture of ethanol and water (20 mL, 70:30 (*v*/*v*)) were ranged from 28.33 to 36.33%. Furthermore, Azman et al. [28] reported that the extraction amount for fresh lemon peels was 11.0%. It appears that modifications to the extraction conditions, including those related to maturity, plant parts, and solvent choice can affect the extract yield results.

### 2.3. Total Phenolic Content

Various investigations have reported that the total phenolic contents (TPC) in foods and plants are related to their antioxidant activities, presumably because of their redox characteristics, which permits them to be active as a hydrogen donor, reducing substances, and/or oxygen singlet quenchers [26]. TPC extracted from the raw materials were expressed in mg GAE equivalents/g of dried weight (DW) (Table 1). Lemon peels showed the highest TPC (162 ± 0.93 mg GAE/g) followed by potato peels (157 ± 0.88 mg GAE/g), olive leaves (62.08 ± 1.12 mg GAE/g), and pomegranate peels (54.84 ± 0.96 mg GAE/g), where lemon peels and potato peels discovered a significant impact on acrylamide formation (Table 1). This demonstrates that the lower the AA formation is, the higher the total phenolic content is. This unveils the considerable effect of total phenolic contents in reducing AA formation in fried potatoes (as discussed below). Comparable results were revealed that the TPC for pomegranate peels was 297.5 mg tannic acid equivalents/g of pomegranate peel extract [26]. Kyriakos et al. [23] reported that the TPC of pomegranate peels has been observed to vary from 18 to 510 mg/g (DW). Moreover, Azman et al. [28] showed that total phenolic content 72.0 ± 0.67 mg GAE equivalents/g for lemon peels which was around 2 times lower than the value reported in this research (162 ± 0.93 mg GAE/g). However, the TPC of our waste samples can be regarded as greater than that found for other agro-industrial byproducts such as grape pomace (31 mg/g DW) [29], and rice hulls (13–26 mg/g DW) [30]. The TPC determined in potato peels extract in our research (157 ± 0.88 mg GAE/g DW) was higher than that found by Mohdaly et al. [31]. These might be due to the drying method adopted since bioactive components are extremely vulnerable to temperature degradation. Furthermore, TPC and the antioxidant potential are influenced by numerous parameters intrinsic and extrinsic such as cultivar, region, sample pretreatment, temperature, storage, light, oxygen and the collection season [32]. Findings of this segment suggested that fruit has a significant amount of phenolic in the outer compartment.

### 2.4. Total Flavonoid Content

Agricultural wastes contain naturally occurring compounds such as flavonoids, hydroxybenzoic acid, anthocyanins, and lignans. Flavonoids possess a broad spectrum of biological and chemical properties such as radical-scavenging activities. Since antioxidant capacity does not always relate to the existence of high amounts of phenolic content, flavonoid content of extracts also needs to be examined [24]. As shown in Table 1, our data revealed a difference in total flavonoids content (TFC) among wastes samples with values 512.62 ± 1.24, 552.03 ± 1.27, 615.48 ± 1.27, and 528.00 ± 1.08 mg QE/g DW in olive leaves, potato peels, pomegranate peels, and lemon peels extracts, respectively. The highest value of TFC was observed in pomegranate peels extracts, which was the lowest TPC, whereas the lowest levels of TFC was found in olive leaves extracts. TFC of the waste lemon peels in our work was higher than that obtained by Azman et al. [28] who found that the TFC of fresh lemon peels from Malaysia was 50.51 mg QE/g. In a recent study, Alexandre et al. [32] studied the TFC of olive leaves of different collection times from Brazil and described that the TFC revealed various levels between autumn and summer, ranging from 513 and 594 mg QE/g. Hsieh et al. [33] quantified a total flavonoid amount of 275 mg/g in potato peels aqueous extracts. Variations could result from several environmental factors including light exposure, temperature, plant cultivar, geographic location and minerals present in the soil, which can affect the synthesis and content of flavonoids.

### 2.5. Antioxidant Scavenging Activity of Various Extracts

Antioxidant activity evaluation depends on the reaction mechanism, so in order to fully assessment the overall antioxidant capability, several assays must be integrated to estimate the antioxidant capacity of extracts [34]. Therefore, in our study, the antioxidant activity in wastes samples has been employed through different methods, including DPPH radical-scavenging activity and ABTS radical scavenging capacity to extend a reliable evaluation of the antioxidant profiles of tested agricultural wastes samples.

#### 2.5.1. DPPH Radical Scavenging Activity

DPPH is scavenged by accepting hydrogen or an electron radical from phenolic bioactive compounds and reduced to 2, 2-diphenyl-1-picrylhydrazine (yellow coloured). DPPH radical scavenging activity was measured in order to evaluate the antioxidant activity of lemon peels, potato peels, pomegranate peels, and olive leaves extracts and the results are shown in Table 1. Results revealed that at 1 mg/mL, the DPPH scavenging activity had decreased in the order of lemon peels (inhibition of 91.80% with EC_50_ 2.60 mg/mL) followed by potato peels (87.80% with EC_50_ 2.77 mg/mL), pomegranate peels (71.40% with EC_50_ 3.10 mg/mL), and then olive leaves, with the lowest inhibitory effect of 20% with EC_50_ 10.85 mg/mL. EC_50_ values express the concentration of antioxidant from the sample needed to reduce its radical concentration by 50%. The lower EC_50_ value represents the higher scavenging activity of a sample. Jeddou et al. [35] exhibit promising outcomes when examining the antioxidant activity of potato peels by various in vitro methods using DPPH (EC_50_ = 2.5 mg/mL) and ABTS radical scavenging activities (14.835 ± 0.1%) [21]. Indeed, it has already been established that lemon peels, potato peels, pomegranate peels, and olive leaves extracts showed strong DPPH scavenging activity as the synthetic antioxidants butylated hydroxytoluene and butylated hydroxyanisole found in study by [21,28], recommending them as potential natural antioxidants. In agreement with our results, Alexandre et al. [32] revealed that olive leaf extracts showed smaller radical scavenging activity collected in October and February (49.94% and 55.5%, respectively). Azman et al. [28] found that extracts of fresh lemon peels showed a high hydrogen donating capacity towards the DPPH radical EC_50_ value (1.30 mg/mL). With regards to one of the other studies, pomegranate peels has showed a slightly lower DPPH inhibition (71.40%) compared to the one mentioned by Shalini et al. [26] who found that the methanol extract of pomegranate peels showed 79.5% antioxidant activity using DPPH assay method. These differences could be caused by the selection of the extraction solvent, which could have an impact on the extraction rate, the method of detection in addition to the variation between fresh and dried peels. In most research, a positive correlation was unveiled between DPPH inhibition value and TPC of various wastes extracts [36]. Our results confirmed this phenomenon, which lemon peels and potato peels showed almost similar phenolic content and antioxidant potential and higher than the other wastes. That means the high level of TPC could be a sign of high antioxidant activity. Chu et al. [37] observed in a study that total flavonoids content in Chinese wild rice contributed more to the antioxidant capacity than total phenolic content. Moreover, Sir Elkhatim et al. [38] observed that the radical scavenging activity (%) of DPPH assay were not significantly correlated with total phenolic or total flavonoids contents, indicating that an increase in the total phenolic content has no effect on DPPH scavenging activity. These observations did not align with our study. It can be concluded that our samples are capable of delaying or halting the oxidation process, thereby providing additional health benefits to consumers.

#### 2.5.2. ABTS Radical Scavenging Activity

ABTS assay is another commonly technique for determination of radical scavenging activity in many foodstuffs. In the presence of samples, the nitrogen atom of ABTS quenches the hydrogen atom in extracts, decreasing the solution colour indicates reduction in ABTS. The evaluated antioxidant activities of lemon peels, potato peels, pomegranate peels, and olive leaves extracts are given in Table 1. All samples minimized the absorbance and the results of ABTS radical scavenging activity of our samples exhibited a similar trend to the results of DPPH assay. A previous study also supports our finding that there was a strong correlation between DPPH and ABTS radical scavenging activities [31]. The ABTS scavenging activity of the samples were in order of lemon peels extract > potato peels extract > pomegranate peels extract > olive leaves extract. In the present study, various samples exhibited a variability in their inhibiting behavior against ABTS radical cation with range of activity from 49.46 ± 0.72 to 94.12 ± 0.48. The higher the ABTS scavenging activity values of the tested samples, the higher the total antioxidants. Lemon peels extract revealed the highest ABTS scavenging capacity. The findings of this study are in agreement with Azman et al. [28] in which the lemon peels extract exhibited high antioxidant activity related to that the citrus peel have a high ascorbic acid content, which could affect their ability to scavenge ABTS radicals. The variations in phenolic contents identified above in our study are compatible with the variations in the antioxidant capacity by ABTS, thus the finding suggests that total phenolic contents are the primary contributors to the antioxidant activity of our tested samples. Antioxidants play an essential role in the mitigation of AA formation by removing the free electron generated during the Maillard reaction. The antioxidants present in the waste extracts reduced the acrylamide concentration (as discussed below).

### 2.6. Identification of Phenolic Compounds of Waste Materials by HPLC Analysis

Phenolics represent the prevalent class of phytochemicals of plants and agricultural wastes. It is evident that the total phenolic compounds estimated by Folin-Ciocalteu assay does not reflect the level or type of phenolic components present in the extracts under study. Therefore, The HPLC analysis of the phenolic compounds in various extracts were employed and gathered in Table 2. HPLC analysis of the various extracts of agricultural wastes revealed a complex mixture of phenolics and flavonoids which are representative of the different structural types. Table 2 reveals that olive leaves extract had the highest amounts of vanillin, ferulic acid, apigenin, quercetin, and coumaric acid (429, 273, 232, 162 and 117 µg/g, respectively). In turn, other authors reported different main compounds in olive leaves, namely, oleuropein and hydroxytyrosol, followed by luteolin-7-O-glucoside [39]. Pomegranate peels extract had the highest amount of gallic acid, catechin, ellagic acid, and methyl gallate (15,329, 5684, 2262 and 267 µg/g, respectively). Kyriakos et al. [23] reported that the main phenolic component found in pomegranate peels is punicalagin (16,670 µg/g followed by gallic, catechin and ellagic acid with a content of 12,580, 8680 and 440 µg/g, respectively. Potato peels extract had the highest chlorogenic acid amount (2067 µg/g) and caffeic acid (270 µg/g); while lemon peels extract had the highest naringenin amount (1881 µg/g). These findings are in agreement with Riciputi et al. [40] who revealed that chlorogenic acid accounted for 49.3–61% of TPC in potato peels with caffeic, ferulic, and catechin as the most prevalent compounds. In lemon peels, the most important phenolic compounds are naringenin and chlorogenic acid, followed by pyrocatechol, rutin and ferulic acid. Other phenolic components have been also identified in lower amount such as gallic acid, caffeic acid, vanillin, methyl gallate, syringic acid, ellagic acid, quercetin, cinnamic acid, apigenin, kaempferol, hesperetin, daidzein, and coumaric acid. The difference in the phenolic compounds profile among the samples in the existing and other studies might be due to many factors including climatic conditions, different extraction conditions, cultivar, fruit maturation, and origin. Additionally, these phenolic compounds, including phenolic acids and flavonoids have exhibited positive effects such as preventing or reducing the risk of AA formation and/or AA precursor formation by trapping intermediates of the Maillard reaction, scavenging free radicals, or by reacting with lipid radicals, which were constant, by the delocalization of unpaired electrons [41].

### 2.7. Effect of Waste Samples on Acrylamide Formation

The effect of agricultural wastes extracts on mitigation of AA formation in processed potato product after frying process in response to pre-treatment with the tested samples was assessed. Four different natural agricultural wastes extracts, all easily obtainable from the wastes of various food industries, were used as potential AA formation inhibitors. The potato slices were soaked in the extracted solutions preparations for 1 h at an ambient temperature, then fried in soybean oil at 180 ± 2 °C for 10 min, the acrylamide was extracts from the samples and quantified by HPLC-DAD. The preliminary analysis of the potato chips (control sampl) revealed the occurrence of acrylamide in potato crisps which was observed to be 104.94 mg/kg (as shown in Figure 1). This concentration is high in comparison with the actual concentration of acrylamide (95.5 mg/kg) in potato chips observed by Marwa et al. [17] for the control potato chips. Following this, the formation of acrylamide was examined as a function of type of the waste, oil uptake, reducing sugar content, and asparagine content, though all these parameters under this research are interlinked. From the acrylamide HPLC chromatogram (Figure 1 and Figure 2), a strong mitigation to the AA formation in potatoes crisps were found with the extracted solutions preparation of the various agricultural wastes. Actual acrylamide levels in potato chips were 69.23, 31.84, 93.31, and 14.58 mg/kg in response to pre-treatment with olive leaves, potato peels, pomegranate peels, and lemon peels extracts, respectively, comparing to the control sample (104.94 mg/kg). Previously, many studies have reported the acrylamide content in fried model system varied widely from 2.26 to over 30.3 μg AA/g fried potato [42]. In another study, the acrylamide concentration in potato crisps ranged from 55 μg/kg to 546 μg/kg [43]. The difference in potato tubers composition even within the same group of foodstuffs, the product, and in the processing procedures used can readily explain the remarked variation. Based on the findings, the quantity of formed acrylamide in potato crisps because of Maillard reactions “non-enzymatic reactions” was strongly reduced by 11–86%, upon immersion the potato slices with the extracted solutions of the agricultural wastes, as a pre-treatment before the frying step. Interestingly, the efficiency of lemon peels extracts for acrylamide mitigation was increased by about 7 folds than control sample, achieving the protective effect of lemon peels on acrylamide formation. As well as this, potato peels have the efficiency to mitigate about 70% of acrylamide in the resulting fried potato (Figure 1). On contrary, pomegranate peels and olive leaves treatments had presented in potatoes crisps a low mitigation of this toxic component, at about 34% and 11%, respectively. This could be correlated with high concentration of total phenolic compounds that lemon peels and potato peels extracts exhibited among the other extracts. In a recent report, Sara et al. [42] studied the effect of polyphenols addition on the AA formation in fried potato crisps and observed that caffeic acid, a single phenolic acid, had AA reduction in the range of 29 to 47%, whereas for tyrosyl acetate, the AA reduction effect was substantially stronger (90%). French fries previously soaked in *Caralluma fimbriata* extract have been also demonstrated to cause a reduction in acrylamide [44]. Furthermore, grape extract has also been found to mitigate the level of acrylamide formation by 60.3% in a potato chip [45]. Other reports investigated the effect of cinnamon, green tea and oregano extracts on the formation of acrylamide in potato crisps, obtaining a mitigation of acrylamide level by 39%, 62% and 17%, respectively [46]. Esfanjani and Jafari [47] demonstrated the addition of various extracts in foods is limited by numerous reasons including low solubility, bioavailability and susceptibility against processing conditions.

It is noteworthy that the inhibitory impact of lemon peels was substantially more effective than that of other three extracts. This outcome might be interpreted because naringenin, as a main component present in lemon peels, is less polar phenolic compound and this actuality appears to be important for a beneficial reduction effect. Our results are supported by a previous study of Napolitano et al. [48] who attributed the reduction in AA formation during frying process to oil polar antioxidants. These substances are able to decrease the oxidation in lipophilic attributed to that their effect take place at the interface between the oil and the polar phase. Moreover, naringenin, the aglycone of naringin, as plant flavonoids are antioxidant constituents of many citrus fruits. Flavonoids linked by glycosidic bonds through sugar ligands such as galactose and glucose, and therefore, they are usually structurally stable at heat treatment. This differs from phenolic acids, which could break down through isomerization via molecular lactone reduction at high temperatures. Furthermore, this variance in AA formation among the various waste extracts might be related to other possible factors such as pH, reducing sugar level, asparagine content, and oil uptake content [49]. Indeed, the wide range of oil uptake in potato crisps (32.50–42.20%) might be behind this great disparity in AA formation among extracts. In general, thermal treatment is known as a considerable factor contributing to the degradation of phenolics, which may lead to a decrease in antioxidant capacity [50].

Potato peels also showed a high impact on the acrylamide formation, and this might be due to its high concentration of chlorogenic acid. Findings obtained in this study confirm earlier results that chlorogenic acid has a beneficial effect on mitigation AA production. It retains a strong redox potency throughout Maillard process, which may restrict the formation of AA [13]. Formerly, chlorogenic acid reduced AA formation in potato powder [51] and in baked potatoes [13]. Furthermore, when kept at an ambient temperature and subjected to light, chlorogenic may relatively converted to caffeic, which could illustrate why caffeic is sometimes considered to be the main phenolic component in some studies [21].

Regarding the quantification and qualification of the phenolic composition of extracts able to improve the reduction effect, it has been reported that the amount and type of phenolic compounds did not always reveal a positive relationship in AA content and, therefore, it did not correlate with the most reduction effect. These results are in line with a previous data by Mildner-Szkudlarz et al. [52], who reported that the addition of quercetin to bread mitigated AA formation but only at low levels, whilst at higher concentrations of quercetin did not show a significant AA mitigation. Sara et al. [42] also confirmed that a nonlinear correlation between concentration of polyphenols and acrylamide formation was observed. Another example was about the use of gallic acid and caffeic acid in two independent studies [53,54] and the results obtained were entirely different: both phenolic acids were non effective to reduce the AA formation in the former experiment but effective in the latter one. Green tea extract is a well-known and potent antioxidant substance because of its high amount of catechin which might be effective in mitigating acrylamide formation, while it did not exhibit an inhibitory effect on the concentration of acrylamide [55].

Despite the fact that phenolic compounds have a propensity to be the highest in antioxidant content, there is beside this, a considerable class of nonphenolic substances that give support to overall antioxidant capacity of food. Different phenolic compounds frequently vary in their ability to act as antioxidants and can interact with other components existent in crude extracts in a cooperative, synergistic or antagonistic way [13]. It is worth mentioning that anti-scavenging activity of the samples was linked to the reduction in AA formation. This may reveal the relatively direct relation of antioxidants in acrylamide formation mitigation.

Thus, taken from the HPLC analysis, the soaking of potato chips in agricultural wastes extracts for 1 h at ambient temperature as pre-treatment before frying in oil abolishes the formation of acrylamide regarding to the control. In this approach, the risks related to the acrylamide consumption could be adjusted and, consequently, controlled.

### 2.8. Effect of L-Asparagine and Reducing Sugars Contents on Reduction in Acrylamide Formation

Different factors including raw ingredients composition and preparation processes could affect the acrylamide formation in foodstuff. In the event that raw materials utilized in food production, various plant types may have an impact on acrylamide formation due to variations in type and content of main acrylamide precursors, i.e., free asparagine and reducing sugars [56]. Potatoes have a high content of asparagine and reducing sugar which are developed to AA formation during frying process. The increased Maillard reaction formation acrylamide in potato strips is mainly due to great amount of reducing sugars (fructose and glucose) and L-asparagine [57]. Therefore, we evaluated the concentration of asparagine and reducing sugars in potato raw material, potato slices after soaking (before frying) in water and in aqueous solutions of different wastes, as well as in potato chips processed (after frying). Compared with the control sample, soaking in the tested wastes extracts effectively reduced not only AA formation level of processed potato chips but also reduced AA precursor contents (asparagine and reducing sugar) in potato slices before frying (Table 3). The reducing sugars and asparagine concentrations in potato raw material used in this study was 690 mg/Kg and 37.70 mg/Kg, respectively. While after soaking in water and in aqueous solutions of olive leaves, potato peels, pomegranate peels, and lemon peels were 442, 417, 492, 492, and 430 mg/Kg for reducing sugars and 26.80, 26.56, 28.44, 29.14, and 26.86 mg/Kg for asparagine content, respectively. The reduction percentage of reducing sugars and asparagine after soaking treatment ranged from 28.70 to 39.57% and from 22.71 to 29.55%, respectively, in the samples. This effect of soaking in water and in aqueous solutions of different wastes might be related to leaching of asparagine and reducing sugars from the surface of potato during soaking. It has been revealed that soaking process reduces the amount of reducing sugars required for Maillard reaction; thereby, mitigating the AA formation level in the end product [58].

The depletion rates of reducing sugars and asparagine after frying process decreased with potato samples soaked in different agricultural wastes and, in consequence, abolished the formation of acrylamide regarding to the control. This may be in part attributed to the hindering reaction of acrylamide precursors by the phenolic compounds in the extracts. Our suggestion seems to be confirmed with the study by Zhu et al. [59] who reported that the reduction in AA formation might be due to the fact that the direct abstraction reaction of AA with bioactive compounds in plant extracts. The decrease in both reducing sugars and asparagine contents after frying was associated by a raise in AA formation in all samples, which is correspondent with the assumption that these compounds are the main precursors of AA formation. Consistent with this finding, Food Drink Europe Acrylamide Toolbox [11], who reported that the AA formation in potato processed is extremely affected by concentration of its precursors, as well as soil type, fertilizers, climate, processing techniques (pre-treatment and the addition of exogenous additives) and storage conditions. From Table 3, it is evident that the depletion rates of reducing sugars were the lowest in lemon peels (43.72%) followed by potato peels (60.98%), olive leaves (69.06%), pomegranate peels (71.14%) and water (74.66%), whereas the highest depletion rates were characteristic for control (86.96%). Whilst the depletion rates of asparagine were the lowest in potato peels (16.74%) followed by lemon peels (29.00%), olive leaves (52.18%), pomegranate peels (54.56%), and water (58.02%), whereas the highest depletion rates were characteristic for control (79.50%). On the other hand, the lowest contained residual reducing sugars and asparagine after frying was control sample, while the most contents were observed for lemon peels and potato peels extracts. In this context, due to high amount of asparagine and reducing sugars in potato raw material, potato crisps (control) exhibit the highest acrylamide level compared with other samples. The presence of these main AA precursors was positively linked with acrylamide contents produced during the frying process. Likewise, Matyas et al. [13] reported that AA contents in potato types were appreciably affected by reducing sugars content. Reducing sugars may be implicated in formation of 5-hydroxymethyl-2-furfural, which could be a potent carbonyl accelerating AA formation during heating in thermally processed foods [60]. Moreover, in research by Matyas et al. [13] and Yang et al. [61] AA concentration in potato chips notably linked with the reducing sugars but not asparagine. The Maillard reaction’s progress in potato is limited more by the content of reducing sugars than by the content of asparagine [62]. Thus, the main acrylamide precursors (reducing sugar and asparagine concentration) in potato chips can be minimized by using soaking of potato slices in the tested agricultural wastes extracts for 1 h at room temperature as a pre-treatment before frying in oil.

### 2.9. Oil Uptake in Treated Potatoes Chips

Oil is absorbed by the food during frying, replacing some of the lost water, is controlled by numerous factors including frying temperature, frying time and oil quality. The oil uptake of potato strips fried at 180 ± 2 °C for 10 min was also measured to examine if there are any variations in the oil absorption in treated potatoes fried due to the various in waste extracts used (Figure 3). Results indicate that the oil uptake in potato fried soaked in the extracted solutions of olive leaves, potato peels, pomegranate peels, and lemon peels extracts was 42.20 ± 0.78, 34.10 ± 0.86, 40.10 ± 0.68, and 32.50 ± 0.90%, respectively. However, the results by Amira et al. [1] revealed that the oil uptake in potato slices with different techniques processing was 9.98–14.9%. The high amount of oil uptake in our study is normal since the potato slices were completely submerged in oil during frying. Moreover, the oil uptake might be also influenced by using higher temperature during frying process. This observation is confirmed by a study conducted by Heeba et al. [63], who considered temperature of 180 °C was more oil imbibition than temperature of 160 °C. The finding also showed that the higher level of oil uptake in potato crisps leading to the higher rate of AA formation and this suggestion confirmed with the study by Amira et al. [1] who reported that a highly significant correlation between oil uptake content and acrylamide formation. Hence, lemon peels with less oil absorption demonstrates more benefits than the other samples in terms of reducing AA formation and oil content uptake, overall calories and employ healthy product. As well as this, consumers today are eating foods and their products with lower fat contents. We hypothesize that the oil penetration in the surface, the heat transfer, moisture evaporation rate and drying at higher temperatures leading to faster AA formation.

### 2.10. Impact on Potatoes Fried Sensory Profile

Sensory analysis of the treated potato fried samples was carried out as it is extremely important to estimate the consumer acceptability of sensorial attributes of the products. A sensory panel evaluated the potatoes fried, taken into observance various descriptors linked to the product color, taste, odor, appearance, texture and overall acceptability. The results of the sensory evaluation show that the potato samples fried soaked in different agricultural wastes extracts as a pre-treatment before frying in oil were remarkably diverse on most attributes compared to control sample (Table 4). According to panelists’ opinion, potato samples fried soaked in both of olive leaves and lemon peels extracts showed an overall acceptability of 8.5 and were higher than control (8.3), while with potato peels extract (8) was bound tightly to the control treatment, whereas potato fried with pomegranate peels extract scored the lowest. The organoleptic properties results evidenced that samples fried with olive leaves, lemon peels and potato peels extracts had the highest scores for taste and color attributes. Moreover, lemon peels and olive leaves extracts were favorable than control in appearance attributes. Fried potato with low fat uptake has a hard and unpleasant texture [64], while our results did not align with this observation. Based on of the sensory analysis results, it can be concluded that samples fried with olive leaves, lemon peels and potato peels extracts were sensorially accepted(Like very much), which encourage to be used as a soaking pre-treatment before frying for their phytochemical properties as antioxidant and inhibiting acrylamide formation potentials to consumers.

## 3. Materials and Methods

### 3.1. Plant Materials

The potatoes (*Solanum tuberosum*) used in this study, belongs to “Hermes” that represents the Egyptian traditional types for fraying (Crisps), were obtained from local market. Fresh pomegranate fruits (*Punica granatum* L.) and fresh lemon fruits (*Citrus lemon*, L.) were obtained from the market, El-Fayoum governorate, Cairo, Egypt, while fresh olive (*Olea europaea* L.) leaves were picked up from farm of El-Fayoum, Egypt. Palm oil was obtained from a restaurant in El-Fayoum, Egypt.

### 3.2. Chemicals and Reagents

Standard acrylamide solution was purchased from BDH electran, England. Acetone, methanol, acetonitrile, and *n*-hexane were obtained from Fisher Chemical, Cornell lab, Egypt; Gallic and formic acids from Fluke Sigma Aldrich; ABTS and DPPH from Sigma Aldrich, USA; Oil (soybean) from local market.

### 3.3. Preparation of Potato Peels, Olive Leaves, Lemon Peels and Pomegranate Peels Powder

Fresh potato and lemon were washed and manually peeled. Olive leaves are washed using tap water to remove any traces of dust. Pomegranate fruits were cut manually to separate peels and seeds. The rind (peels) and the cleaned olive leaves thus obtained, cut into small pieces and dried in a hot air oven at 50 ± 5 °C for 6 h. The dried pieces were cooled, ground to a fine powder in a laboratory disc mill (Type: Braun KM 32, Waiblingen, Germany). The powdered material that passed through a 60-mesh screen sieve were packed in high density polyethylene bags and kept at room temperature (25 ± 5 °C) until further use.

### 3.4. Preparation of Waste Extracts

Ten grams of each peel and olive leaves powder were soaked in 100 mL Methanol 80% for 48 h at ambient temperature, subsequently mixture was filtered on (Whatman No. 1) and the residues were subject to extraction again in the same condition. The collected filtrate extracts were evaporated from the solvent at 40 °C by using rotary evaporator. Finally, the extracts were kept in the freezer and the percentage of the yield of extracts has been calculated.

### 3.5. Chemical Composition of the Waste Extracts

Moisture content was estimated by the air oven assay according to 44-15A, AACC [65]. Protein content was determined by the Kjeldahl method [66]. The total ash content was determined according to AOAC 972.15 [65]. The crude fat content was obtained by exhaustively extracting 2 g of each sample in a Soxhlet using petroleum ether [20]. All of the samples were analyzed in triplicate. Carbohydrate content was calculated by differential method.

### 3.6. Determination of Waste Materials Total Phenolic Compounds

The quantification of total phenolic compounds (TPC) was determined using Folin-Ciocalteu reagent and according to the method of Kumar et al. [67]. TPC of the extracts was calculated using standard curves (0–600 mg/L) of gallic acid. Data were expressed as (mg GAE/g).

### 3.7. Determination of Total Flavonoids Compounds of Waste Materials

Total flavonoids contents (TFC) were determined using method of Ebrahimzadeh et al. [68]. The absorbance of the reacting mixture was measured at 415 nm using T80 UV–vis spectrophotometer (Leicestershire LE17 5BH, Alma Park, Wibtoft, United Kingdom). The calibration curve was obtained for quercetin as a standard using the same steps. Finally, the TFC were computed as (mg QE/g).

### 3.8. Determination of Antioxidant Activity of Waste Materials

#### 3.8.1. DPPH Radical Scavenging Assay

DPPH radical assay was determined to estimate the efficiency of the studied wastes antioxidant capacity according to the method described by Mohdaly and Mohamed, [69]. The working solution of DPPH was prepared by mixing 45 mL of methanol with 10 mL of DPPH stock solution. Briefly, 1 mL of each prepared test sample was added to 1 mL of DPPH working solution, and a control that consists of 1 mL DPPH working solution with methanol (1 mL) was also prepared. Then, mixtures were mixed and incubated for 30 min at ambient temperature in the dark. The absorbance at 515 nm against a blank was recorded using a T80 UV–Vis spectrophotometer (Leicestershire LE17 5BH, United Kingdom). The percentage of DPPH radical scavenging activity of the extract samples was calculated as follow:DPPH Radical scavenging activity (%) = [(AC − AS)/AC] × 100
where AC: the absorbance of the control; AS: the absorbance of the sample.

The data were also expressed as EC_50_ of DPPH (sample concentration able to reduce DPPH radical by 50%).

#### 3.8.2. ABTS Radical Scavenging Assay

The 2,2′ -azino-bis-3-ethylbenzthioazoline-6-sulfonic acid (ABTS) scavenging activity of the samples were assayed according to a previously reported method of Mohdaly et al. [70]. Briefly, the ABTS radical cation working solution was obtained by reaction of potassium persulphate (2.45 mM) with a stock solution of ABTS (7 mM), stored in the dark at ambient temperature for 12 h. The mixture was then diluted with alkaline phosphate buffer to achieve an absorbance of 0.700 at 734 nm. To determine the ABTS radical cation scavenging activity, each prepared sample (0.2 mL) was added to the obtained working solution of ABTS (0.8 mL). After 6 min in the dark, using a UV–Vis spectrophotometer the absorbance of the mixture was recorded at 734 nm. ABTS scavenging activity was calculated as follow:ABTS Radical scavenging activity (%) = [(A_0_ − A)/A_0_] × 100

A_0_, the absorbance of control; A, the absorbance of sample.

### 3.9. Identification of Phenolic Compounds of Waste Materials by HPLC Analysis

HPLC analysis was carried out using an Agilent 1260 series with Eclipse C18 column (4.6 mm × 250 mm i.d., 5 μm). Mobile phase solvent consisted of (A) water and (B) 0.05% trifluoroacetic acid in acetonitrile with flow rate 0.9 mL/min. The sequential programming of the mobile phase in a linear gradient was as follows: 0 min (82% A); 0–5 min (80% A); 5–8 min (60% A); 8–12 min (60% A); 12–15 min (82% A); 15–16 min (82% A) and 16–20 min (82% A). The detector was monitored at 280 nm. The injection volume was 5 μL for each sample.

### 3.10. Fried Potato Chip Processing

Potatoes were obtained from local market and washed under tap water. The washed potato tubers were peeled manually, cut into potato chips slices of approximately 2.0 mm in thickness using potato chips maker. The peels of pomegranate, lemon, potatoes, and olive leaves powder were soaked in distilled water at a rate of 10 g/L for a night. Then, the chips slices were soaked in the aqueous extracts of the different waste peels at room temperature for an hour before frying; meanwhile, the control samples were submerged in water. Potato chips slices were drained for 2 min before being fried. After that, the chips slices of each variety were fried separately in deep-frying, which was carried out in soybean oil at 180 ± 2 °C for 10 min in a regular Tefal pan. Directly after frying, samples of potato fried crisps were cooled on an absorbing paper (fat absorption). Afterwards, a part of the batch was homogenized for acrylamide analysis, while the other part was used for sensory analysis.

### 3.11. Acrylamide Determination

#### 3.11.1. Sample Preparation

QuEChERS method was used to determine the acrylamide concentrations in potato fried and the extraction procuders were done according to the method of Bellah and Nelson [71]. Approximately 1 g homogenized potato fried crisps were weighed into a 50-mL centrifuge tube, then 10 mL of n-Hexane was added, and the solution was placed on a shaker for 1 min. Subsequently, 10 mL of both distilled water and acetonitrile were added, followed by adding of Bond Elute QuEChERS extraction salt for acrylamide. The tubes were immediately shaken for 1 min again and centrifuged for 5 min at 4000 rpm.

#### 3.11.2. Dispersive SPE Cleanup

From the acetonitrile layer, 6 mL were transferred into a Bond Elute QuEChERS Dispersive SPE 15 mL tube. Then, tubes were centrifuged for 5 min at 4000 rpm. A 1000-μL amount of samples was taken in an autosampler vial to estimate the acrylamide concentration by HPLC-DAD analysis.

### 3.12. Determination of Reducing Sugars

Reducing sugars were estimated by schooler assay as proposed in the AOAC [72]. Ten grams of potato samples were weighed and 40 mL of 80% alcohol was added. The extraction was carried out for 1 h. in a water bath at 80 °C, followed by centrifugation. Next, the resulting residue was subjected to extraction again in the same condition. The two collected extracts were mixed, and 80 alcohol was added to a final volume of 100 mL.

### 3.13. Determination of Asparagine

Asparagine was analyzed by using HPLC according to Igor et al. [73]. A total of 100 mg of the sample was added with 5 mL H_2_O and 5 mL of HCl (6 M), followed by filtration. Then, 1 mL of the filtrate was dried and resuspended in (0.1 M) HCl and injected into HPLC. The mobile phase consisted of buffer Sodium phosphate buffer pH 7.8 (A) and ACN: MeOH: H_2_O 45:45:10 (B) at a flow rate 1.5 mL/min. The DAD was monitored at 338 nm (Bandwidth 10 nm).

### 3.14. Oil Uptake Content

Oil uptake content of potato chips were analyzed by Soxhlet method, according to the method described by AOAC [72].

### 3.15. Sensory Analysis of Potato Chips

In order to estimate the acceptability of our studied samples with different processing conditions, sensory evaluation was performed. Organoleptic attributes including color, taste, odor, appearance, texture and overall acceptability of fried samples were evaluated by a semi-trained panel of the department of food science and technology, Fayoum university (as regular consumers) of 10 members with ages ranging from 25 to 55. Samples were evaluated according to the nine-point hedonic scale started from (9 to 1) which; 9: Like extremely, 8: Like very much, 7: Like moderately 6: Like slightly, 5: Neither like nor dislike, 4: Dislike slightly 3: Dislike moderately, 2: Dislike very much, 1: Dislike extremely.

### 3.16. Statistical Analysis

Analysis was carried out in triplicate and the data were presented as the mean ± standard deviation for all samples. Results were processed by Excel.

## 4. Conclusions

Acrylamide is a carcinogenic compound present in potato fried and its existence needs to be reduced. Hence, four different natural agricultural wastes extracts used as a soaking pre-treatment method before frying have been proposed to reduce acrylamide formation. Total phenolic, total flavonoids content, HPLC analysis of phenolic compounds, and antioxidant scavenging activity of tested samples were determined. In addition to acrylamide, reducing sugar, asparagine, and oil uptake were also measured. The experimental results showed that lemon peels potato peels, olive leaves, and pomegranate peels had considerable amounts of TPC, TFC and high scavenging activity. Thus, the pre-treatment potatoes with the aforementioned wastes succeeded to mitigate AA formation in all samples (86.11%, 69.66%, 34.03%, and 11.08% for lemon peels, potato peels, olive leaves, and pomegranate peels, respectively). Emphasizing the potential link between total phenolic amount, antioxidant activity of samples as well as oil uptake, L-asparagine and reducing sugars contents in the controlling of acrylamide formation. It is critical to note that a greater reduction in this harmful substance was seen in lemon peels, whilst lower reduction in acrylamide formation was characteristic for pomegranate peels. It is clear that this work has demonstrated a healthier and a favorable of fried potato wherein the antioxidant efficiency of waste extracts provided a reduction in the amount of acrylamide. Furthermore, lemon peels and potato peels rendered potato chips more favorable, with low AA concentration and minimal oil absorption.

## Figures and Tables

**Figure 1 molecules-27-07516-f001:**
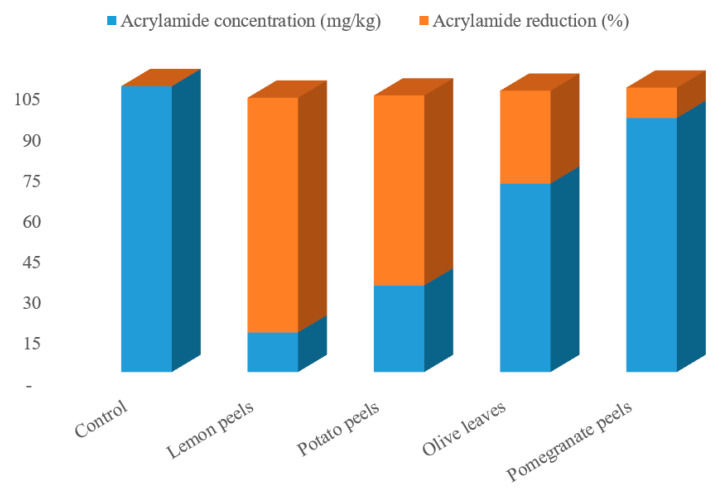
Acrylamide contents and percentage of acrylamide reduction in various fried potato chips treated with different agro-industrial extracts.

**Figure 2 molecules-27-07516-f002:**
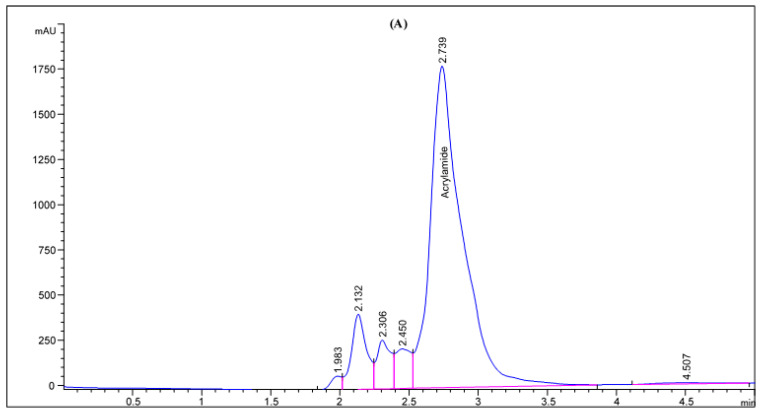
HPLC chromatogram of acrylamide in various fried potato chips (**A**) control; (**B**) soaked in pomegranate peels extract; (**C**) soaked in olive leaves extract; (**D**) soaked in potato peels extract; (**E**) soaked in lemon peels extract.

**Figure 3 molecules-27-07516-f003:**
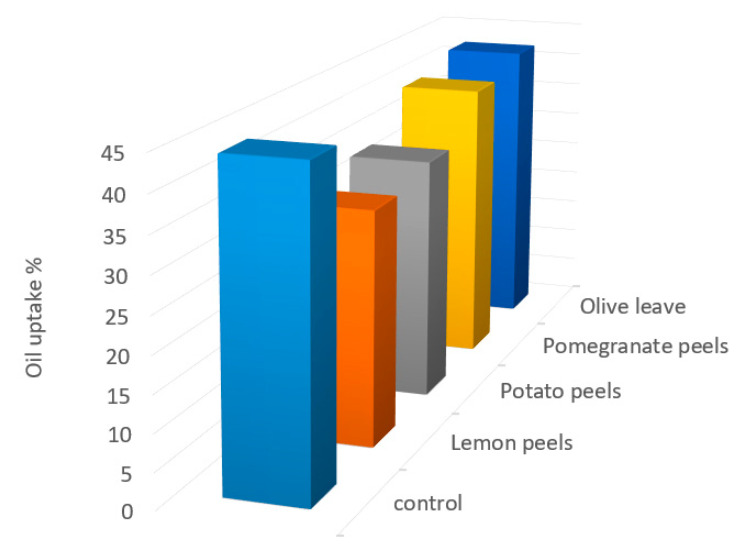
The oil uptake content (%) in control and treated potatoes fried with different agro-industrial extracts.

**Table 1 molecules-27-07516-t001:** Proximate analysis *, yield extract, TPC, TFC, and antioxidant scavenging activity of lemon peels, potato peels, olive leaves, and pomegranate peels samples.

Parameter	Lemon Peels	Potato Peels	Pomegranate Peels	Olive Leaves
**Moisture**	73.00 ± 0.22	82.00 ± 0.17	74.00 ± 0.19	59.00 ± 0.54
**Ash**	7.42 ± 0.28	4.34 ± 0.34	3.03 ± 0.82	5.85 ± 0.44
**Crude fat**	3.70 ± 0.24	2.07 ± 0.32	2.50 ± 0.46	4.20 ± 0.23
**Crude protein**	1.89 ± 0.22	2.33 ± 0.38	1.41 ± 0.40	2.28 ± 0.44
**Carbohydrate ****	13.99 ± 1.32	9.26 ± 0.96	19.05 ± 1.12	28.67 ± 1.41
**Yield extract %** **(DW** **)**	24.85 ± 1.93	14.50 ± 1.14	63.29 ± 0.83	32.98 ± 0.94
**TPC mg GAE/g (DW** **)**	162 ± 0.93	157 ± 0.88	54.84 ± 0.96	62.08 ± 1.12
**TFC (mg QE/g DW)**	528.00 ±1.08	552.03 ± 1.27	615.48 ± 1.27	512.62 ± 1.24
**DPPH Radical scavenging activity (%)**	91.80 ± 0.88	87.80 ± 1.08	71.40 ± 0.74	20 ± 1.32
**EC_50_ (mg/mL)**	2.60 ± 1.88	2.77 ± 1.24	3.10 ± 1.08	10.85 ± 1.32
**ABTS Radical scavenging activity (%)**	94.12 ± 0.48	90.82 ± 0.80	78.24 ± 0.68	49.46 ± 0.72

* Based on wet weight (%). ** Total carbohydrates calculated by difference. TPC: total phenolic content; TFC: total flavonoid content; DW: dry weight. Data are mean ± SD of three replicates.

**Table 2 molecules-27-07516-t002:** HPLC analysis of phenolic compounds in various extracts samples (µg/g).

Phenolic Compound	Lemon Peels	Potato Peels	Olive Leaves	Pomegranate Peels
Gallic acid	82.08	67.83	157.44	15,329.66
Chlorogenic acid	1614.21	2067.63	nd	633.86
Catechin	nd	169.81	293.92	5684.66
Methyl gallate	13.86	0.99	6.84	267.33
Caffeic acid	20.11	270.28	6.03	43.97
Syringic acid	67.06	21.28	70.26	82.34
Pyrocatechol	531.67	nd	nd	nd
Rutin	355.32	9.36	73.78	nd
Ellagic acid	205.73	44.23	53.06	2262.34
Coumaric acid	7.76	1.55	117.31	nd
Vanillin	89.06	4.23	429.68	48.45
Ferulic acid	224.42	61.70	273.28	146.89
Naringenin	1881.80	31.83	147.99	442.38
Daidzein	210.64	26.48	nd	7.21
Querectin	35.17	1.18	162.41	93.02
Cinnamic acid	1.08	2.24	33.95	nd
Apigenin	2.83	nd	232.42	nd
Kaempferol	12.42	nd	nd	nd
Hesperetin	20.50	nd	nd	nd

nd = Not detected.

**Table 3 molecules-27-07516-t003:** Changes in reducing sugar and asparagine contents in potato slices during soaking and frying of potato chips.

	Reducing Sugars Content (mg/Kg)				Asparagine Content (mg/Kg)			
Treatment	After Soaking	After Frying	Reduction % after Soaking	Depletion Rates after Frying	After Soaking	After Frying	Reduction % after Soaking	Depletion Rates after Frying
**Control**	690 ± 0.86	90 ± 0.97	00.00	86.96%	37.70 ± 0.76	7.73 ± 0.82	00.00	79.50
**Water**	442 ± 0.76	112 ± 0.88	35.94	74.66%	26.80 ± 0.96	11.25 ± 0.88	28.91	58.02
**Lemon peels**	430 ± 0.88	242 ± 1.46	37.68	43.72%	26.86 ± 0.98	19.07 ± 1.22	28.75	29.00
**Potato peels**	492 ± 0.96	192 ± 0.89	28.70	60.98%	28.44 ± 0.86	23.68 ± 1.18	24.56	16.74
**Pomegranate peels**	492 ± 0.83	142 ± 0.86	28.70	71.14%	29.14 ± 1.13	13.24 ± 0.98	22.71	54.56
**Olive leaves**	417 ± 1.08	129 ± 1.13	39.57	69.06%	26.56 ± 1.18	12.70 ± 1.24	29.55	52.18

Data are mean ± SD of three replicates.

**Table 4 molecules-27-07516-t004:** Sensory profile of control and treated potato fried samples.

Parameter	Control	Lemon Peels	Potato Peels	Pomegranate Peels	Olive Leaves
**Color**	8.4 ± 0.70	8.7 ± 0.48	8.5 ± 0.79	5.6 ± 0.97	8.5 ± 0.53
**Taste**	8.0 ± 0.94	8.6 ± 0.52	8.1 ± 1.32	6.8 ± 0.92	8.5 ± 0.71
**Odor**	8.1 ± 1.29	7.5 ± 1.08	8.1 ± 1.29	7.5 ± 1.08	8.3 ± 0.67
**Appearance**	8.5 ± 0.53	8.6 ± 0.52	8.4 ± 0.70	6.0 ± 1.01	8.6 ± 0.52
**Texture**	8.4 ± 0.70	8.2 ± 0.63	8.5 ± 0.71	7.6 ± 1.26	8.7 ± 0.48
**Overall acceptability**	8.3 ± 0.82	8.5 ± 0.53	8.0 ± 1.05	6.3 ± 0.67	8.5 ± 0.53

Data represented in means of ten panelists’ ± standard deviation.

## Data Availability

All data generated and analysed during this study are included in this manuscript.

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
