# Peer review of "Potential of Low Cost Agro-Industrial Wastes as a Natural Antioxidant on Carcinogenic Acrylamide Formation in Potato Fried Chips"

_molecules, 2022, doi:10.3390/molecules27217516_

Round 1

Reviewer 1 Report

This is an interesting approach to the use of waste extracts as sources of antioxidants. Regarding the content and the capacity of the phenolic compounds of these extracts to reduce the generation of acrylamide, it has been extensively studied previously.

It is not necessary to include the chemical composition of the primary metabolites of the raw materials, since their methanolic extracts are only used, and the study focuses on their phenolic content.

With respect to extraction yield, it should be included if they are with respect to fresh or dry weight and the solvent it has been carried out in table 1.

Figure 2 should be included in supporting information.

Some typographic mistakes:

Line 70: usuful, varous

Line 71: solanum tuberesum

Line 82:phytochmical

Line 433: poatao

Line: 538: Tuberosum

Line 719: 2020

Reviewer 2 Report

This paper was designed to investigate the effect of natural wastes such as potato peels, olive leaves, lemon peels and pomegranate peels extracts on acrylamide formation in potato fried chips. The author did a lot of work to achieve their objective. However, some modifications should be performed.

1.      Line 26, it should be “reduce” instead of “abolish”.

2.      In Table 1, carbohydrate measurements should show standard deviation data.

3.      In Table 1, compared with other samples, pomegranate peels has the lowest total phenolic content but the highest total flavonoid content. Can the author explain why this result appears?

4.      Why did the author choose to soak the peel and olive peel in 80% methanol for 48 hours during the preparation of waste extract?

5.      Line 616, please add the corresponding unit after“16-20”.

6.      3.16 Statistical analysis section. Please add a specific data analysis method.

Round 2

Reviewer 2 Report

The authors have revised their manuscript accordingly.